# Isoliquiritigenin Protects Neuronal Cells against Glutamate Excitotoxicity

**DOI:** 10.3390/membranes12111052

**Published:** 2022-10-27

**Authors:** Arina Zgodova, Svetlana Pavlova, Anastasia Nekrasova, Dmitriy Boyarkin, Vsevolod Pinelis, Alexander Surin, Zanda Bakaeva

**Affiliations:** 1Laboratory of Neurobiology and Fundamentals of Brain Development, National Medical Research Center of Children’s Health, 119991 Moscow, Russia; 2Department of Psychiatry and Psychosomatics, Sechenov First Moscow State Medical University (Sechenov University), 119146 Moscow, Russia; 3Department of Pharmacology, Clinical Pharmacology and Biochemistry, Chuvash State University named after I.N. Ulyanov, 428015 Cheboksary, Russia; 4Institute of General Pathology and Pathophysiology, Russian Academy of Medical Sciences, 125315 Moscow, Russia; 5Department of Medicine, General Biology and Physiology, Kalmyk State University named after B.B. Gorodovikov, 358000 Elista, Russia

**Keywords:** isoliquiritigenin, neuroglial culture, neuroprotection, [Ca^2+^]_i_, mitochondrial potential, NMDAR, OCR

## Abstract

It is considered that glutamate excitotoxicity may be a major factor in the pathological death of neurons and mediate the development of neurodegenerative diseases in humans. Here, we show that isoliquiritigenin (ILG) at a concentration of 0.5–5 µM protects primary neuroglial cell culture from glutamate-induced death (glutamate 100 µM). ILG (1 µM) prevented a sharp increase in [Ca^2+^]_i_ and a decrease in mitochondrial potential (ΔΨm). With the background action of ILG (1–5 µM), there was an increase in oxygen consumption rate (OCR) in response to glutamate, as well as in reserve respiration. The neuroprotective effect of ILG (5 µM) was accompanied by an increase in non-mitochondrial respiration. The results show that ILG can protect cortical neurons from death by preventing the development of calcium deregulation and limiting mitochondrial dysfunction caused by a high dose of glutamate. We hypothesize that ILG will be useful in drug development for the prevention or treatment of neurodegenerative diseases accompanied by glutamate excitotoxicity.

## 1. Introduction

Glutamate is an endogenous excitatory neurotransmitter in the mammalian central nervous system, which in high doses may be a major contributor to pathological cell death within the nervous system and may be involved in many acute and chronic brain diseases [1,2], including stroke, traumatic brain injury, and Alzheimer’s disease [3,4,5]. Glutamate excitotoxicity is also the central mechanism of long-term neuronal death and delayed effects induced by traumatic brain injury [6,7]. Glutamate excitotoxicity is associated with a dramatic increase in intracellular free calcium concentration ([Ca^2+^]_i_) mediated by prolonged activation of N-methyl-D-aspartate receptors (NMDARs) [8,9,10]. Then, the inevitable processes occur sequentially, starting with a loss of mitochondrial membrane potential (ΔΨm) [8,11], and release and translocation of AIF (from mitochondria into the cytosol, such as Bcl-2 and Bax), which regulate apoptotic programmed cell death [12,13], followed by generation of reactive oxygen species (ROS) and breakdown of intracellular organelles [14,15]. The intracellular changes that arise during this process include the activation of Ca^2+^-dependent enzymes, for example, proteases, with subsequent changes in the structure and activity of many proteins both in the cytosol and the organelles [11,16]. In addition, an increase in glutamate concentration leads to the inversion of the astrocyte glutamate transporter [17] and impaired glutamate uptake by presynaptic neurons [18]. This vicious circle leads to the mass death of neurons. In this regard, the search for neuroprotective agents against glutamate excitotoxicity is highly relevant. Traditionally neuronal cultures serve as models for the research and development of biologically active substances with neuroprotective properties. At the same time, it is possible to use the analysis of such parameters as [Ca^2+^]_i_ and ΔΨm [19,20], as well as the oxygen consumption rate (OCR) [21,22] for this purpose.

Isoliquiritigenin (ILG) refers to flavonoids with chalcone structures originating from Leguminosae family plants (e.g., Glycyrrhizae radix). This class of polyphenolic substances are plant secondary metabolites that have attracted scientific attention due to their different health-promoting effects, including neuroprotective activity [23]. 

The neuroprotective mechanisms of flavonoids can be manifest at different levels. The importance of flavonoids in neurological disorders is due to blood-brain barrier penetration. However, most dietary polyphenols show a low oral bioavailability due to presystemic elimination during absorption [24]. High amounts of dietary polyphenols remain unabsorbed from the gastrointestinal tract, modulating the gut microbiota and intestinal barrier function to establish the so-called gut–brain axis [25]. On the other hand, a widely discussed mechanism is the antioxidant property of polyphenols. Acting as antioxidants, flavonoids including ILG can improve neuronal cell survival in different toxicity [26]. However, it has become clear that the mechanisms of action of polyphenols go beyond the modulation of oxidative stress [27]. The majority of the experimental data indicate that flavonoid compounds regulate gene transcription mediated by phosphorylation in signaling cascades, due to the effect on the protein kinase activity [28]. The anti-inflammatory and immunomodulatory benefits of various classes of flavonoids are obtained through regulating key signaling pathways (NF-κB, MAPK, JAK/STAT, etc.) in different cells involved in neuroinflammation [23].

The exposure of neuronal cells to toxic concentrations of glutamate initiates a signaling cascade that mediates cell death through excessive reactive oxygen species production and mitochondrial damage. We hypothesized that glutamate neurotoxicity and pathogenic events including altered signaling pathways and the generation of free oxygen radicals triggered by the increased Ca^2+^ theoretically can be inhibited by ILG. ILG-mediated neuroprotection. It has been demonstrated in the mouse hippocampal cell line HT-22 [13,29,30,31]. In this model, ILG reversed reactive oxygen species production and mitochondrial depolarization, as well as glutamate-induced changes in the expression of the pro- and anti-apoptotic proteins [13]. Besides, *in vitro* ILG attenuated mitochondrial membrane potential loss in neuronal pheochromocytoma cell models of oxidative and nitrosative stress [32]. Moreover, ILG is thought to be an NMDA receptor antagonist that was demonstrated using primary culture rat cortical neurons [33].

ILG has been shown to have neuroprotective effects in several models. However, the mechanisms of ILG action in relation to mitochondrial functions still remain to be established. The aim of the presented study was to evaluate the effect of ILG on the physiological function of mitochondria and its potential neuroprotective properties in glutamate-induced toxicity in the primary culture of rat cortical neurons.

## 2. Materials and Methods

### 2.1. Cortical Neuroglial Cell Culture Preparation

Neuroglial cultures were prepared from the cerebral cortex of Wistar rat pups (P1-P2) by the standard method [6]. Briefly, the rats were anesthetized, decapitated, and the cortex was removed and separated from the meninges. The extracted tissues were washed with a Ca^2+^- and Mg^2+^-free Hank’s solution, dissected, incubated in a papain solution (10 min, 37 °C) and dissociated by pipetting. After sedimentation in a centrifuge (three times, 200 g, 5 min) cells free of debris were washed with a Ca^2+^-containing solution and in a neurobasal medium (NBM, Gibco, Waltham, MA, USA) at the end. Homogeneous cells were resuspended to a concentration of 10^6^ cells/mL in the neurobasal medium (NBM) supplemented with B-27 Supplement and penicillin/streptomycin; 200 µM aliquots of the cell suspension were transferred into wells of a 48-well plastic plate (Costar, Glendale, AZ, USA), 250 µL aliquots were transferred onto coverslips attached to the wells of 35-mm glass-bottom Petri dishes (MatTeck, Ashland, MA, USA), or, alternatively, 100 µL aliquots were transferred into wells of 24-well Seahorse plate (Agilent Technologies, Santa Clara, CA, USA). The cells were kept in an incubator at 37 °C in an atmosphere of 5% CO_2_/95% air and a relative humidity of 100%. Cytosine arabinoside (5 µM) was added to the medium every 2–3 days to prevent the proliferation of glial cells and to obtain cultures with a percentage of neurons ≥90%. The cells were supplied with nutrients every three days by replacing 1/3 of the medium with a fresh one. The cultures were used in experiments 9–11 days after plating (9–11 days in vitro, DIV). 

Experiments with animals were performed in accordance with the ethical principles and regulatory documents recommended by the European Convention on the Protection of Vertebrate Animals used for experiments (Guide for the Animals and Eighth Edition. 2010), as well as in accordance with the “Good Laboratory Rules practice”, approved by order of the Ministry of Health of the Russian Federation No. 199n of 04.01.2016. All the protocols were approved by the Ethics Committees at the National Medical Research Center for children’s health, Russian Ministry of Health.

### 2.2. Evaluation of Neuronal Viability

The survival of neuroglial cultures under conditions of glutamate excitotoxicity was determined by a biochemical method 24 hours after the glutamate (100 µM, Gly 10 µM, –Mg^2+^, 1 hour) exposure. The water-soluble form of tetrazolium (Water Soluble Tetrazolium, WST, Roche Diagnostics GmbH, Mannheim, Germany) is especially convenient for assaying the quantification of viable cells, conversion of WST to formazan reflects the activity of hydrogenases; therefore, it is cleaved to form a formazan dye only by metabolically active cells [34]. The optical density of the WST was measured using a ClarioStar multimodal plate reader (BMG Labtech, Ortenberg, Germany). The wavelength for measuring the absorbance of the formazan product was 440 nm, and the reference wavelength was 700 nm. The obtained data were normalized by taking the absorbance of formazan in the control cultures as 100% and the absorbance in the wells containing only buffer as 0%. ILG was added 1-hour prior to glutamate, and its exposure was maintained during the rest of the experiment. 

### 2.3. Measuring of [Ca^2+^]_i_ and ΔΨm in Cortical Neurons 

The fluorescence-microscopic measurements of the intracellular free Ca^2+^ concentration ([Ca^2+^]_i_) and the mitochondrial transmembrane potential (ΔΨm) were performed in individual cells using the fluorescence-microscopic analysis system based on an Olympus XI-70 microscope (Japan), equipped with a 175 W xenon lamp, 20x/NA = 0.75 fluorite objective, a Sutter Lambda 10-2 filter wheels controller (Sutter Instruments, Novato, CA, United States), and a CoolSnap HQ2 camera (Photometrics, Tucson, Arizona, United States). 

Measurements of [Ca^2+^]_i_ were performed using the low-affinity fluorescent Ca^2+^ indicator Fura-FF (excitation 340 and 380 nm, emission 525 nm). Changes in [Ca^2+^]_i_ are presented as a ratio of fluorescence intensities, excited at 340 and 380 nm (F340/F380). Stock solutions of Fura-FF/AM (2 mM in dimethylsulfoxide) preliminary mixed with nonionic detergent Pluronic F-127 were added to cells in 1 mL of NBM to final concentrations 4 µM and 0.02%, respectively (60 min, 37 °C). For simultaneous monitoring of the [Ca^2+^]_i_ and ΔΨm changes, cells were loaded with a potential-sensitive dye rhodamine 123 (ex/em 485/525 nm, 6.6 µM, 15 min, 37 °C). Fura-FF signals were recorded at 30 s time-lapse. Measurements were performed at 27–29 °C in buffered saline containing 130 mM NaCl, 5 mM KCl, 2 mM CaCl_2_, 1 mM MgCl_2_, 20 mM HEPES, and 5 mM glucose (pH 7.4). Glutamate was washed out using a buffer in which Ca^2+^ was replaced with EGTA (100 mM) and Mg^2+^ (2 mM). The maximal Fura-FF signals were determined using the Ca^2+^ ionophore ionomycin (2 µM) in the presence of 5 mM CaCl_2_ (without Mg^2+^) at the very end of the experiments. The amount of Ca^2+^ stored in mitochondria was estimated with depolarizing the organelles with the protonophore carbonyl cyanide 4-(trifluoromethoxy) phenylhydrazone (FCCP, 1 µM) in nominally Ca^2+^-free NB. All solutions were prepared on the day the experiments were done. Data were recorded with the MetaFluor software (Molecular Device, San Jose, CA, USA) as images of the entire observation area. The recorded images were processed using the MetaFluor Analyst software (Molecular Device Corp., United States). The principle of data calculation and the parameters were described and shown before [6,35].

### 2.4. Measurements of the Neuroglial Culture Oxygen Consumption Rate

A standard mito-stress test was used in experiments in which the effects of glutamate and ILG on the oxygen consumption rate (OCR, pmol/min) of cortical culture were evaluated. OCR was measured using the Seahorse XFe24 Extracellular Flux Analyzer (Agilent Technologies, CA, USA), at 37 °C, in an assay medium, consisting of 130 mM NaCl, 5 mM KCl, 2 mM CaCl_2_, 1 mM MgCl_2_, 20 mM HEPES, 5 mM Glucose, at pH ∼7.4. The microplate-based respirometry utilizes a 24-well plate format and quantifies the OCR at different times, following the addition of ILG (5, 1, 0.5, and 0.1 µM), glutamate and glycine (100 µM and 10 µM, respectively).

Prior to each experiment, neurons in each well plate were washed twice with 500 µL of the medium. Four wells per plate did not contain neurons, serving as “blank” wells, to control temperature-sensitive fluctuations in O_2_-sensitive fluorophore emission. Following washing, each well was filled with 525–600 µL of the medium, and the plates were placed in a CO_2_-free incubator (37 °C) for 1 hour before each set of measurements to further purge CO_2_ and to allow temperature and pH equilibration. The plates were then loaded into the XF24 analyzer, and the sensors had been being additionally calibrated for 15 min before the first measurement was done. 

The substances of interest prepared in an assay medium (in volume 75 µL) in accordance with the Seahorse XF Cell Mito Stress Test Kit, were preloaded into reagent delivery chambers (A–D) and injected at predesignated intervals. ILG or glutamate was injected through port A, oligomycin was used as the second injection (port B) at a concentration of 1 µM. The respiratory chain uncoupler, carbonyl cyanide p-(trifluoromethoxy) phenylhydrazone (FCCP, 1 µM), was added through port C. A mixture of inhibitors rotenone (0.5 µM), and antimycin A (0.5 µM) were added through port D. In another series of experiments ILG, glutamate, FCCP and rotenone/antimycin A were added sequentially through ports A, B, C and D, respectively. The respirometry cycle consisted of a 3-min medium mix, a 1-min wait, and a 3-min measurement stage (one speed point). Subsequently, the obtained data were calculated via the Seahorse XF Cell Mito Stress Test Report Generator, which automatically calculates and reports assay parameters. 

Spare respiratory capacity was determined as described before [36]. The non-mitochondrial oxygen consumption rate is the minimum OCR measured after the antimycin A/rotenone injection. Basal respiration is the difference between the OCR before the application of the first agent and the non-mitochondrial oxygen consumption rate. Maximal respiration is the difference between the OCR amplitude observed in the presence of FCCP and the non-mitochondrial oxygen consumption rate. The spare respiratory capacity is the difference between maximal and basal respiration. All data were normalized to the basal level of the respiration rate (OCR just before the application of any agent, 100%).

### 2.5. Statistical Analysis

Data were statistically processed using the GraphPad Prism 8.0 software (GraphPad Software Inc., San Diego, CA, USA). The data were checked for normality using three tests: D’Agostino–Pearson omnibus normality test, the Shapiro–Wilk normality test, and the KS normality test. In the case of normal data distribution, differences were estimated using the One-Way ANOVA with Sidaks multiple comparisons test; otherwise, the Kruskal–Wallis with Dunn’s multiple comparisons tests were used.

## 3. Results

### 3.1. Measuring of Neuronal Viability by Biochemical WST-Test

Using the WST-test to assess cell survival, we demonstrated that glutamate (100 µM) reduced viability by 32%, consistent with our earlier studies [21]. ILG (0.5 and 1 µM) significantly increased cell survival up to 90 ± 22% (*p* < 0.001), 83.4 ± 15.5% (*p* < 0.0001) and 81 ± 13% (*p* < 0.05), respectively, in the condition of glutamate excitotoxicity (Figure 1). The drug at higher concentrations (10–100 µM) did not have a significant effect compared to glutamate alone. There was no significant difference between the groups. Сell cultures incubated with low concentrations of ILG up to 10 µM demonstrated higher WST-test absorbance levels compared to the control cultures. In the presence of higher concentrations (50 µM and 100 µM) cell death increased by about 26% and 27%, respectively, in comparison with intact cells from the control group.

### 3.2. Measuring of [Ca^2+^]_i_ and ΔΨm in Cortical Neurons 

Measurements of [Ca^2+^]_i_ and ΔΨm were performed in cortical neurons with the dose of ILG, which proved to be the most effective one in dose-dependence cell viability experiments. Fluorescence microscopy measurements showed that incubation with ILG (1μM, 15 min) did not change [Ca^2+^]_i_ in resting neurons (Figure 2A). The addition of glutamate (100 µM, 10 µM Gly, 0 Mg^2+^, 15 min) induced the development of delayed calcium deregulation (DCD) and synchronous changes in [Ca^2+^]_i_ and ∆Ψm (Figure 2B). Similar synchronism of [Ca^2+^]_i_ growth and ∆Ψm was repeatedly observed under the action of toxic doses of glutamate on cultured neurons [8,11,37].

It was indicated that DCD in сells after the administration of ILG (1 μM) increases slower in response to glutamate exposure than DCD in the cells from the control culture so the lag phase of the onset of DCD is observed to be prolonged (Figure 2A,E). An important parameter which reflects the dynamics of DCD is the area under the curve after the DCD onset of glutamate action since it reflects the process of Ca^2+^ entry into the cytoplasm during the activation of glutamate receptors [6]. This parameter was significantly higher in the neurons from the control cultures than in neurons in cultures where ILG was applied (Figure 2F). The other parameter such as tg α of Fura-FF fluorescence intensity at the EGTA treatment, showed a decrease in [Ca^2+^]_i_ during its energy-dependent removal [6], conversely, it demonstrated weaker “washing-out” of Ca^2+^ in the culture with ILG(Figure 2G).

In order to assess the contribution of mitochondria to the calcium-dependent processes in parallel with the assessment of the dynamics of [Ca^2+^]_i_, we measured the mitochondrial potential (ΔΨm). The recovery rate of ΔΨm during the “wash-out” period (administration of EGTA) was estimated with the slope of the graph reflecting the change in Rh123. The sharper slope is associated with a faster mitochondrial potential recovery rate. There were unidirectional changes in the integrated fluorescent response both for Rh123 and Fura-FF during the application of EGTA solution (Figure 2B,C). The similar synchronous changes of the [Ca^2+^]_i_ and ΔΨm were indicated before [6,38] which confirms the participation of mitochondria in glutamate-induced changes in calcium homeostasis. However, the rate of mitochondrial potential recovery in neurons with ILG 1 μM was significantly lower than in neurons in intact cultures (Figure 2H). It means cells with ILG 1 μM need more time to reach a full recovery of mitochondrial potential.

### 3.3. Measurements of the Neuroglial Culture Oxygen Consumption Rate (OCR)

In line with the determination of ΔΨm, the integral marker of the mitochondrial functional state improvement is the rate of oxygen consumption [36,39]. Figure 3A represents the change of OCR over time (min) in neuroglial cells culture treated with ILG 0.1–5 μM measured in a standard mito-stress test experiment. We determined that 5 μM ILG, caused a slight increase in OCR over time (min) after its injection. Figure 3B represents the same mito-stress test performed with glutamate. Experimental data showed that the sequential addition of glutamate and oligomicyn caused inhibition of mitochondrial respiration, which was expressed in the absence of a response to the FCCP addition. Therefore, in the following series of experiments, we evaluated the effects of different doses of ILG on OCR with neuroglial cells under conditions of glutamate excitotoxicityte without oligomycin (Figure 4A).

As shown in Figure 4B, ILG at low doses (0.1–1 μM) did not influence basal oxygen consumption. The test of the application of the highest dose, 5 μM ILG, caused a decrease in basal respiration (Figure 4B) and a slight increase in overall OCR (Figure 4C). Further analysis of the results clarified that this effect was associated with an increase in non-mitochondrial respiration, and uptake rate in ILG before glutamate injection (Figure 4C). This tendency was observed on all concentration levels, but only 5 µM ILG showed a significant effect (*p* < 0.05). With the addition of glutamate (100 μM, Gly 10 μM), cells initially responded with an increase in OCR, which decreased over time (Figure 4A). However, in the presence of ILG 5 μM the cells maintained higher oxygen uptake (Figure 4D). To evaluate the spare respiratory capacity (SRC) of mitochondria during glutamate application, the potent uncoupler of mitochondrial oxidative phosphorylation, FCCP (1 μM), was added. A significant increase in SRC was observed only at 1 and 5 μM concentrations of ILG (Figure 4E).

## 4. Discussion 

Excitatory neurotransmitter glutamate production leading to overstimulation of the NMDA receptors is a factor in neuronal damage in traumatic brain injury, in strike and in various chronic neurodegenerative diseases [3,4,5]. The neurotoxicity of glutamate in high concentrations is associated with Ca^2+^ intense entry into neurons and with the development of DCD and synchronous profound mitochondrial depolarization [8,39,40]. Mitochondria play a central role in neuron survival and death: they produce ATP, sequester Ca^2+^, generate reactive oxygen species, and undergo Ca^2+^-dependent permeabilization of the inner membrane [37,41,42]. 

Our attention was drawn to studies [30] where ILG prevented the increase in [Ca^2+^]_i_ accompanied by activation of calcineurin in glutamate-treated HT22 cells. The authors demonstrated that ILG protects against glutamate-induced mitochondrial fission with inhibiting the increase of mitochondrial ROS and intracellular calcium, which are accompanied by dephosphorylation of Drp1 (Ser637), and consequently attenuates glutamate-induced neuronal cell death [30]. Another study which focuses on at the neuroprotective effects of the traditional Japanese medication Yokukansan (10–300 µg/ml), which is composed of seven active drugs was considered. In the NMDA receptor binding and receptor-linked Ca^2+^ influx assays, ILG attached to NMDA receptors and inhibited the glutamate-induced increase in Ca^2+^ influx [33]. 

In the present research we used rats’ neuroglial cortical cell cultures. It was unique in ILG investigations that [Ca^2+^]_i_ and ΔΨm measurements at the same time were carried out at the same time in individual neurons exposed to a neurotoxic concentration of glutamate. This allowed a more in-depth study of the neuroprotective mechanism of ILG. We performed a series of dose-dependence cell viability experiments. Survival analysis during which we used biochemical tests showed that ILG in doses of 0.5–5 μM significantly increased the level of living cells in the condition of glutamate excitotoxicity. At higher concentrations (50 and 100 μM) ILG alone revealed significant neurotoxicity (Figure 1). Next, we used the safest yet effective dose of 1 μM for studying mitochondrial functions and Ca^2+^-homeostasis to elucidate the protective mechanism of ILG against glutamate neurotoxicity. Statistical analysis showed differences in the distribution of the data form, namely, against the background of the action of ILG, the proportion of cells with a large lag value increased (Figure 2E). This means that calcium deregulation in neurons incubated with ILG occurred much later. The area under the glutamate and EGTA curves are the important parameters reflecting the dynamics of DCD. These parameters reflect, respectively, the process of Ca^2+^ entry into the cytoplasm during the activation of glutamate receptors and, conversely, a decrease in its cytoplasmic concentration during the energy-dependent removal of Ca^2+^ carried out with plasma and mitochondrial transporters [6]. The integrated fluorescent response reflects the change in fluorescence intensity of Fura-FF in cells during the 15 min action of glutamate (Figure 2F). The integral fluorescence intensity of Fura-FF during glutamate exposure was lower (*p* < 0.001) in the presence of ILG (Figure 2F), which means [Ca^2+^]_i_ level is decreased compared to control cultures. There were synchronous changes in each individual neuron in the integrated fluorescent response both for Rh123 and Fura-FF during the application of EGTA solution (Figure 2B,C). The unidirectional changes of the [Ca^2+^]_i_ and ΔΨm have been observed before [6,21,35]. However, untypical effects for substances with neuroprotective properties were also observed. The rate of mitochondrial potential recovery during the post-glutamate period in neurons in buffers containing ILG (1μM) was lower (*p* < 0.001) than in control cultures (Figure 2H). Additionally, a difference between groups was observed in [Ca^2+^]_i_ at the EGTA application in the post-glutamate period (Figure 2G). As it is known, plasma membrane Ca^2+^-ATPase actively participates in the regulation of intracellular Ca^2+^ in post-glutamate time by extruding Ca^2+^ outside the cell [43,44]. Some studies indicate that many natural flavonoids inhibit plasma membrane Ca^2+^-ATPase with different potency, which could be caused by the different mechanisms not yet explored for ILG [45]. 

The research of bioenergetic processes was the next step of our study. The standard mito-stress test with oligomycin showed that ILG had no effect on the basic parameters of respiration (Figure 3A), including mitochondrial ATP-linked respiration and proton leak (not shown). However, we observed that ILG at high dose which maintained the OCR at a constant level after FCCP injection and until rotenone/antimicyn A was added (Figure 3A). Increased spare capacity was also observed under conditions of glutamate excitotoxicity (Figure 4E). In response to glutamate, cells with ILG (1 and 5 μM) exposure maintained a higher oxygen consumption rate. At the same time, ILG at 5 μM increased non-mitochondrial respiration in the presence of glutamate. Non-mitochondrial respiration can also be changed by glutamate excitotoxicity [46,47]. Recently, research demonstrated that the exposure of neuronal cells to toxic concentrations of glutamate initiates a signaling cascade, which mediates cell death via excessive reactive oxygen species production and mitochondrial damage [48]. Flavonoids including ILG can improve neuronal cell survival in different toxicity due to their antioxidant property [26]. So, glutamate neurotoxicity pathogenic events including altered signaling pathways, and the generation of free oxygen radicals triggered by the increased Ca^2+^ theoretically can be inhibited with ILG. On the other hand current data indicates that ILG triggers intracellular non-mitochondrial reactions that use oxygen as a substrate. These processes, in turn, do not affect the functioning of mitochondria but help to maintain spare capacity under the condition of glutamate excitotoxicity. We considered the involvement of oxygen-consuming reactions, such as those involving cytosolic oxidases (NOX, MAO, HO, etc.), could be affected with ILG and searched for corresponding literature.

Extra-mitochondrial respiratory burst is classically linked to the NOX family of nicotinamide adenine dinucleotide phosphate oxidases of phagocytes. However, there is increasing evidence that not only in microglia (brain phagocytes) contains NOX enzymes are they but also expressed in neurons, astrocytes, and the neurovascular system. There is proof that NOX2, rather than mitochondria, is the primary source of NMDAR-induced superoxide production in cultured cortical and hippocampal neurons [49,50,51,52] and NOX2 suppression has been shown to prevent excitotoxic cell death [50,51]. It has been demonstrated that mitochondrial antioxidants inhibit reactive oxygen species production by mitochondria and reduce NADPH oxidase activity [53]. Flavonoid substances [54] and namely ILG [55], due to their membranotropic nature, can prevent the either activation or inhibition of NOX. Additionally, most of the chalcones showed potent MAO-B and MAO-A suppression [56] and reduction of dopaminergic neurodegeneration and psychostimulant-induced toxicity [57]; this data cannot explain the increase in extramitochondrial oxygen consumption which occurred in our experiment with the ILG. 

Probably, the ILG-mediated increase in the extramitochondrial oxygen consumption demonstrated in the study could be associated with the expression of the activity of other oxidases. One possible enzyme that can be expressed with ILG is heme oxidases (HO). Different HO isoforms were found in the nervous system [58]. HO-1 is a heat shock protein family member that metabolizes heme to biliverdin (converted into bilirubin), carbon dioxide and ferrous iron [59]. Bilirubin has been recently described as an endogenous antioxidant, which can prevent cell damage mediated by ROS, as well as nitric oxide. The protective actions of HO-1 include metabolizing potentially toxic heme. Additionally, HO-1 metabolizes heme to CO, which acts on cGMP for vasodilation. As described before [60], ILG and some other flavonoids [61,62] increase cytosolic HO-1 levels and reduce oxidative stress. We suppose that HO-1 could be activated with ILG application in glutamate neuronal excitotoxicity. This issue will be the focus of the following studies. 

## 5. Conclusions

In this study, the effects of ILG on the neuronal survival and homeostasis of Ca^2+^ in cultured neuronal cells during glutamate excitotoxicity were investigated. Research suggest that ILG concentrations of 1–5 µM reduce the early [Ca^2+^]_i_ spike in response to glutamate which can be estimated by measuring [Ca^2+^]_i_ in individual neurons. Аt the same time, ILG attenuates mitochondrial membrane potential loss, thus affecting the functional state of mitochondria, which play a crucial role in the injury and death of neurons. The neuroprotective effect of ILG is accompanied by non-mitochondrial respiration, which means that ILG can maintain a higher oxygen consumption rate and spare capacity in response to glutamate exposure. These results may contribute to the development of new therapeutic drugs based on ILG (1–5 µM) for neurodegenerative disorders related to glutamate toxicity.

## Figures and Tables

**Figure 1 membranes-12-01052-f001:**
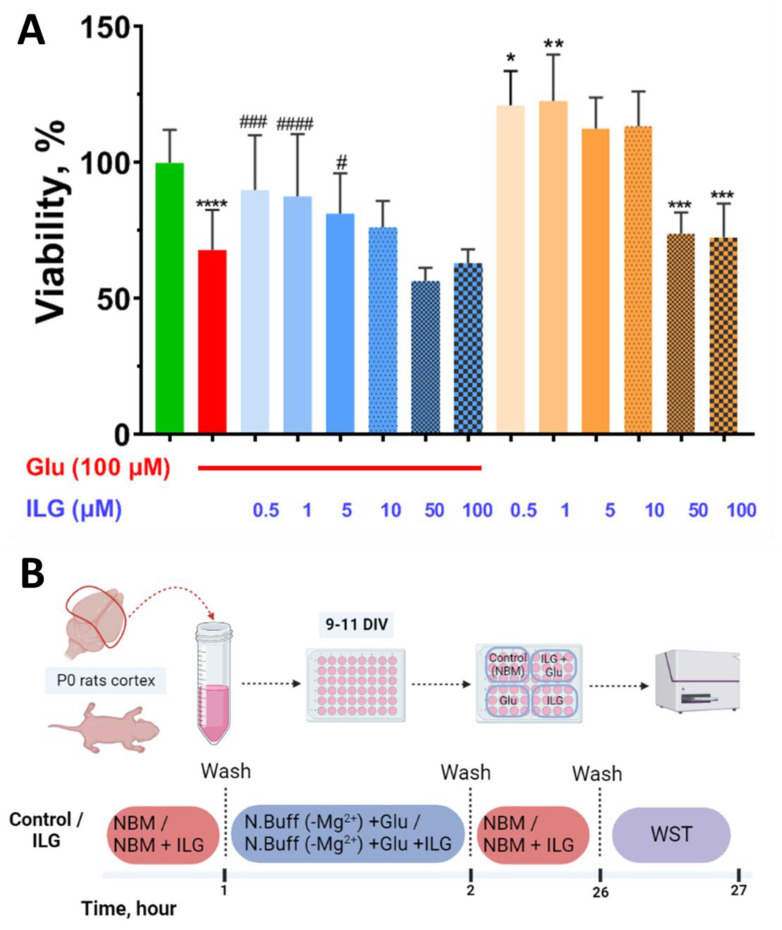
Effect of ILG (0.5–100 µM) on the survival of neuroglial culture cells under conditions of glutamate excitotoxicity (WST test). (**A**) Average viability in groups, determined using the WST test. (**B**) Scheme of experiments with a sequence of drug additions and timeline. Data are represented as *M*
*±*
*SEM*. Five experiments were performed (4–8 wells per group in an experiment). * *p* < 0.05 compared to control; ** *p* < 0.01 compared to control; *** *p* < 0.001 compared to control; **** *p* < 0.0001 compared to control; # *p* < 0.05 compared to glutamate; ### *p* < 0.001 compared to glutamate; #### *p* < 0.0001 compared to glutamate.

**Figure 2 membranes-12-01052-f002:**
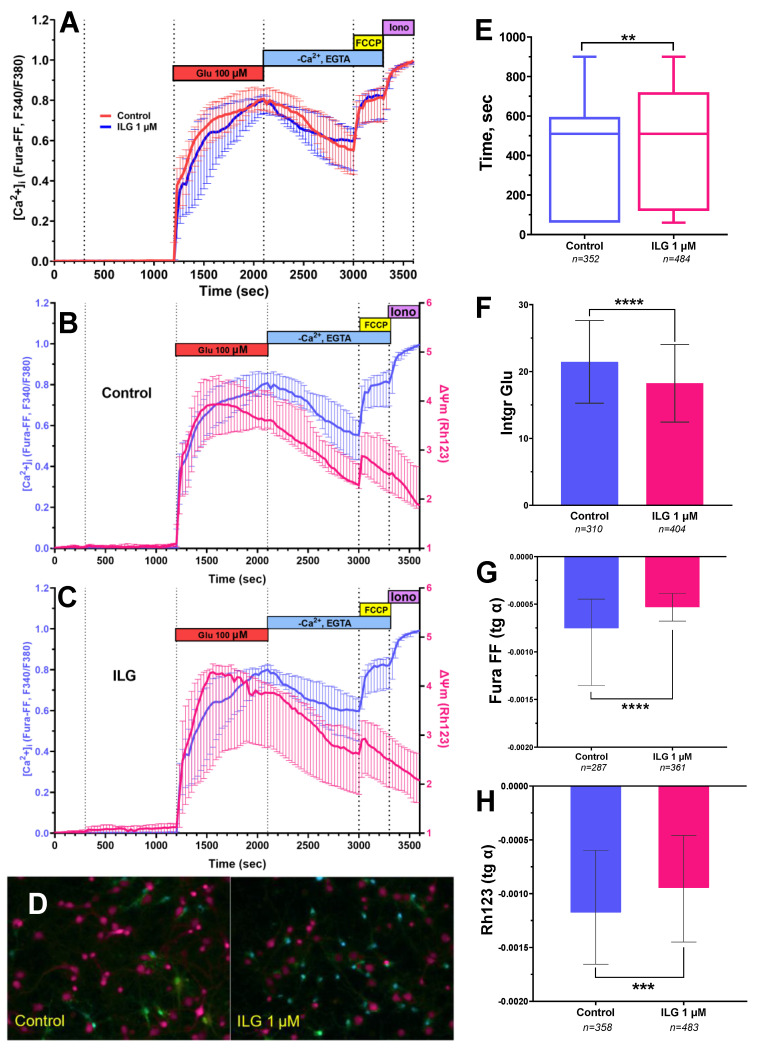
Effect of ILG on the calcium homeostasis and mitochondrial potential (ΔΨm) in primary neuroglial culture cells under conditions of glutamate excitotoxicity. ** *p* < 0.005, *** *p* < 0.001, **** *p* < 0.0001. (**A**) Averaged dynamics of Fura-FF fluorescent signal changes during the experiment are presented as Median with IQR. (**B**) Synchronous changes in [Ca^2+^]_i_ and ΔΨm measured with calcium indicator Fura-FF and a potential sensitive probe Rh123, respectively, in control cell cultures, and (**C**) during ILG exposure. Changes in [Ca^2+^]_i_ are presented as ratios of fluorescent Ca^2+^ indicator Fura-FF signals measured at 340 and 380 nm excitation (F340/F380) and recorded at 525 nm. In each neuron, the F340/F380 ratio is normalized to the baseline value at rest. Changes in Rh123 fluorescence (excitation: 485 nm; emission: 525 nm) are presented as the ratio F/Fo, where F is the current fluorescence intensity, and Fo is its value at the beginning of the experiment. Curves are presented as Median ± CI. (**D**) Fluorescent images of cell cultures after glutamate addition in control group and in group with ILG exposure from representative experiments are presented. (**E**) Lag-phase before the onset of delayed calcium deregulation. (**F**) The integrated fluorescent response reflects the change in fluorescence intensity of Fura-FF in cells during the 15 min action of glutamate or glutamate with 1 μM ILG. (**G**) The slopes (tg α) of linear approximations of the signal curve of the potential-sensitive fluorescent dye Fura-FF during application of EGTA solution. (**H**) The slopes (tg α) of linear approximations of the signal curve of the potential-sensitive fluorescent dye Rh123 during application of EGTA solution. There were 6 different experimental cell cultures. *n*—number of cells.

**Figure 3 membranes-12-01052-f003:**
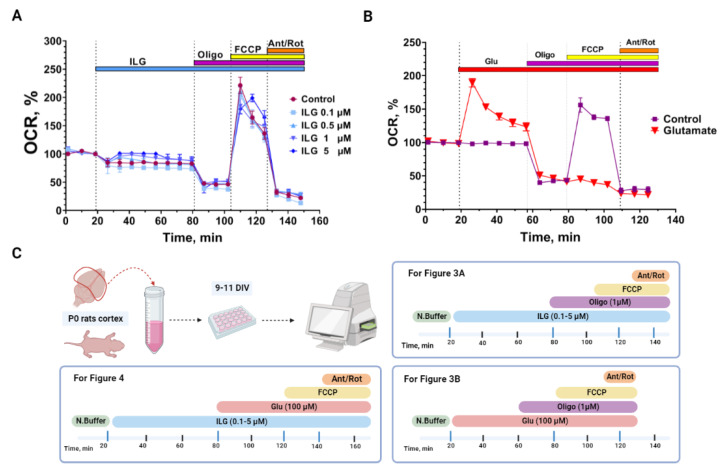
The results of standard mito-stress test performed with ILG/glutamate and oligomicyn. (**A**) Cumulative curve of ILG effects on OCR. (**B**) Cumulative curve of Glu effects on OCR. Figure 1. (**C**) Scheme of experiments with sequence of injection and timeline for Figure 3 and Figure 4. 3 experiments were performed for each mito-stress test design. All data were normalized to the basal OCR of the cumulative curve (Mean ± SEM).

**Figure 4 membranes-12-01052-f004:**
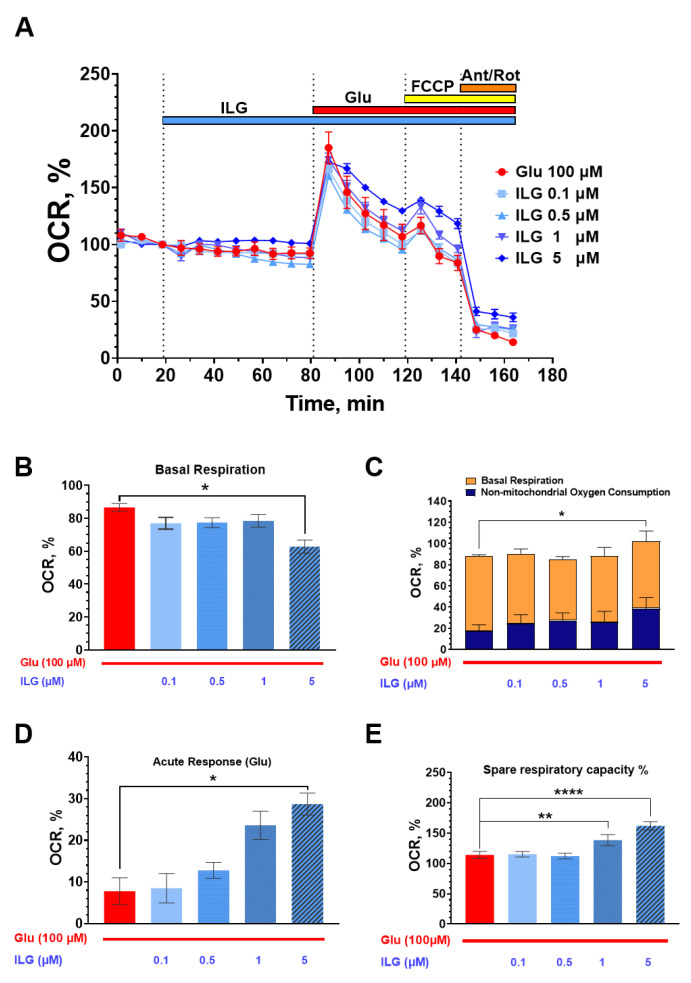
Effect of glutamate and ILG on the oxygen consumption rate (OCR) in primary neuroglial culture. (**A**) Cumulative curve of the effect of glutamate on OCR in the presence of ILG; (**B**) Basal mitochondrial respiration. (**C**) Oxygen uptake rate in ILG before glutamate injection (divided into mitochondrial and non-mitochondrial uptake rates). (**D**) Acute response to glutamate (corresponds to 118 min since experiment started). (**E**) Spare respiratory capacity of mitochondria. Three experiments were performed for each mito-stress test design. All data were normalized to the basal OCR of the cumulative curve (Mean ± SEM). * *p* < 0.05; ** *p* < 0.01; **** *p* < 0.0001.

## Data Availability

Not applicable.

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
