# Peer review of "Isoliquiritigenin Protects Neuronal Cells against Glutamate Excitotoxicity"

_membranes, 2022, doi:10.3390/membranes12111052_

Round 1

Reviewer 1 Report

The manuscript by Zgodova et al. studies the neuroprotective role of Isoliquiritigenin against glutamate excitotoxicity. The manuscript reports the neuroprotective effect through assaying cell viability, intracellular calcium levels, mitochondrial membrane potential, and mitochondrial respiration. Although the results are interesting, the experimental setup and data representation lack rigor.

Major comments

1.      The OCR experiments are not conducted properly; there are major problems with the experimental design.

a.      The basal respiration calculation is flawed as the cells are not treated with Glutamate during the initial phase when basal respiration is calculated. Why was the glutamate treated at 80 mins?

b.      Why are there no untreated controls in the Seahorse experiment?

c.      Why was Oligomycin treatment not performed? How was mitochondrial ATP-linked respiration and proton leak evaluated?

2.      The calcium measurements are not convincing due to the lack of response curves and images. In Figure 2 (A), bar graphs and mean values are different. Indirect Mitochondrial calcium measurements after FCCP treatment is mentioned in the methods but not shown in the manuscript. 

Author Response

The OCR experiments are not conducted properly; there are major problems with the experimental design.

The basal respiration calculation is flawed as the cells are not treated with Glutamate during the initial phase when basal respiration is calculated. Why was the glutamate treated at 80 mins?

Thank you for the valuable comment! We did a standard experiment with Oligomycin (Figure 3). We evaluated the effect of ILG on the basal respiration by calculation according to the manufacturer's protocol as the difference between the last value before oligomycin injection and non-mitochondrial respiration value (i.e. after blockade of the 1st and 3rd complexes of the respiratory chain) (Schuh et al. 2011; Divakaruni et al., 2014). The calculation showed that ILG not affected on the basal respiration in contrast to glutamate (not shown).

In another series of experiments we added all the studied substances successively through the injection ports. This allowed us to further calculate such parameters as the acute response to glutamate and to ILG. Basal respiration was calculated as the difference between the point before the addition of glutamate and the point after addition respiratory chain inhibitors  (Ant/Rot).  Glutamate was added 60 minutes after the injection of ILG, so that ILG could show its maximum effect. Unfortunately, the system Seahors does not involve drug washing-out. For a better understanding, we have added a scheme of experiments with sequence of injection and timeline (Figure 3С).

Why are there no untreated controls in the Seahorse experiment?

Figures 3A and 3B show curves for untreated controls.

Why was Oligomycin treatment not performed? How was mitochondrial ATP-linked respiration and proton leak evaluated?

Thanks for the very valuable note! On the recommendation of the Reviewer, we did a standard experiment with Oligomycin. The corresponding curves have been inserted (Figure 3A and 3B). Our measurements showed that ILG had no effect on ATP-linked respiration and proton leakage.  As you can see on the graph 3B, oligomycin added after glutamate in our experiments completely blocked respiration without the possibility of its recovery. The similar influence of Glu on OCR has been described previously (Grebenik et al., 2020; Belosludtsev et al., 2021; Bakaeva et al., 2022). Therefore, we decided not to use oligomycin in the glutamate model.  Further additions of FCCP and antimycin with rotenone did not make any difference in respiration. We concluded that supplementation of protonophore (FCCP) and respiratory chain inhibitor combinations (Ant/Rot) should be retained as they allow calculation of reserve and maximum respiration. Corresponding changes were made in the Figures 3 and 4 and in the text.

The calcium measurements are not convincing due to the lack of response curves and images.

Dynamic response curves and representative images were added.

In Figure 2 (A), bar graphs and mean values are different.

Statistical analysis showed differences in the distribution of the data form in the absence of a difference in medians, namely, against the background of the action of ILG, the proportion of cells with a large Lag value increased. We have changed the diagram type for better clarity.

Indirect Mitochondrial calcium measurements after FCCP treatment is mentioned in the methods but not shown in the manuscript.

Measurement of calcium concentration directly in the matrix of mitochondria of neurons subjected to the excitotoxic action of glutamate is methodologically difficult, since synthetic calcium indicators created for mitochondria (Rhod-2 and its analogs) are not selective enough and their fraction that remains in the cytosol masks the fluorescence of the indicator in mitochondria. Genetically encoded calcium sensors are expressed with very low efficiency in terminally differentiated cells, which include neurons. Therefore, we applied an “indirect” method for determining the amount of Ca2+ in mitochondria. This method is based on the fact that mitochondrial depolarization leads to the release of Ca2+, which is retained in mitochondria as long as there is a potential difference on the inner mitochondrial membrane (Abramov et al., 2008). The amplitude of Ca2+ rise in the cytosol in response to mitochondrial depolarization, for example, with the help of a protonophore FCCP, can serve as a measure of mitochondrial calcium capacity (Belosludtsev et al., 2021).

References

Schuh, Rosemary A. et al. 2011. “Adaptation of Microplate-Based Respirometry for Hippocampal Slices and Analysis of Respiratory Capacity.” Journal of Neuroscience Research 89(12): 1979–88.

Divakaruni, Ajit S. et al. 2014. 547 Methods in Enzymology Analysis and Interpretation of Microplate-Based Oxygen Consumption and PH Data. 1st ed. Elsevier Inc. http://dx.doi.org/10.1016/B978-0-12-801415-8.00016-3.

Belosludtsev, Konstantin N. et al. 2021. “The Effect of DS16570511, a New Inhibitor of Mitochondrial Calcium Uniporter, on Calcium Homeostasis, Metabolism, and Functional State of Cultured Cortical Neurons and Isolated Brain Mitochondria.” Biochimica et Biophysica Acta - General Subjects 1865(5): 129847. https://doi.org/10.1016/j.bbagen.2021.129847.

Grebenik, E. A. et al. 2020. “Chitosan-g-Oligo(L,L-Lactide) Copolymer Hydrogel for Nervous Tissue Regeneration in Glutamate Excitotoxicity: In Vitro Feasibility Evaluation.” Biomedical Materials (Bristol) 15(1).

Krasil’nikova, Irina et al. 2019. “Insulin Protects Cortical Neurons Against Glutamate Excitotoxicity.” Frontiers in Neuroscience 13(September): 1–12.

Abramov AY, Duchen MR. Mechanisms underlying the loss of mitochondrial membrane potential in glutamate excitotoxicity. Biochim Biophys Acta. 2008 Jul-Aug;1777(7-8):953-64. doi: 10.1016/j.bbabio.2008.04.017. Epub 2008 Apr 18. PMID: 18471431.

Reviewer 2 Report

This study has generated some interesting data regarding the neuroprotective effects of isoliquiritigenin (ILG) in a neuronal glutamate excitotoxicity injury model.  The study examined the effects of ILG on neuronal survival, calcium influx and several mitochondrial parameters after glutamate exposure which would be of value to researchers interested in neuroprotection.

Overall, written quality of manuscript could be improved with copyediting to improving wording and sentence structure.  In addition, Results and Discussion needs significant improvements.  Other suggestions to improve manuscript are provided below.

I have made no attempts to improve wording or sentence structure.

Suggestions to improve manuscript.

Can I suggest changing “Glu” to “glutamate” throughout text in manuscript.

Results section needs to be improved to better follow and understand results obtained in study (especially Results summarised in Figure 2); see comments below.

Discussion also needs to be improved and simplified to better explain the implications of the results obtained in the study.

Line 132:  Was ILG present during the 1-hour exposure of neurons to glutamate?  Please provide more details regarding glutamate exposure protocol.

Line 176: please provide details of assay medium.

In Culture Oxygen Consumption Rate (OCR) studies were neuronal cultures pre-treated with ILG for 1-hour before glutamate exposure?  Please better describe sequence of substance additions and time frames for OCR studies.  Why was glutamate concentration reduced to 30µM in the OCR studies?  A diagrammatic representation of timeline procedures/steps for OCR studies could be useful.

Results

Line 205: change “0,5” to “0.5”.  Please use decimal point rather than “,” throughout manuscript.

Line 206: The 5 µM ILG concentration also appeared to significantly improve neuronal survival after glutamate treatment.

Figure 1; What do 22%, 20% and 13% above bars for 0.5, 1 and 5 µM ILG concentrations represent???

Line 210:  Suggest rewording “a high activity of dehydrogenases” “.. higher WST-test absorbance levels..”

Line 210/211: “There were no significant differences between the groups.”  What are the authors referring to here???

Line 223/224: “Fluorescence microscopy measurements showed that incubation with ILG (1μM, 15 min) did not change [Ca2+]i in resting neurons.”   Where is data/Figure showing this???

Line 224/225: “The addition of Glu (100 μM, 10 μM Gly, 0 Mg2+, 15 min) induced development of delayed calcium deregulation (DCD) and synchronous changes in [Ca2+]i and ΔΨm.”  Where is data/Figure showing this???

Lines 226/227: Similar synchronism of [Ca2+]i growth and ΔΨm was repeatedly observed under the action of toxic doses of Glu on cultured neurons [8, 11, 37].” Where is data/Figure showing this???  Why are references included here???

Figure 2:  Should graphs A-D also show results for glutamate treatment only????  Or is Control the glutamate treatment (if so then control, no glutamate needs to be shown)

Figure 2: “ B) Integral fluorescent response 232 (Glu). (C) The slopes (tg α) of linear approximations of the signal curve of the potential-sensitive fluorescent dye Fura-FF during application of EGTA solution. (D) The slopes (tg α) of linear approximations of the signal curve of the potential-sensitive fluorescent dye Rh123 during application of EGTA solution.”   Should this be defined in Materials and Methods rather than in Results section.

Lines 238-239:  Glutamate response with respect to DCD in Figure 2A needs to be shown.  Also, result with respect to DCD (Fig 2A) looks the same for Control and ILG.

Lines/paragraph 251-261; Could all this be simplified.  Aren’t results in Fig 2C summarised in the paragraph above??

Also sentence “However, the rate of mitochondrial potential recovery in neurons with ILG 1μM was significantly lower than in neurons of intact cultures (Figure 2D).” be improved.  Do you mean the time for mitochondrial potential recovery significantly shorter???

Figure 3B: “As shown in Figure 3B, ILG at low doses (0.1–1 μM) did not influence basal oxygen consumption”. This is with glutamate exposure (30µM) correct?, but at what time point after glutamate exposure or duration of glutamate exposure????

Looking at Fig 3B, it would appear 5 µM causes decrease OCR.

Figure 3D:  What was time frame for measurements?

Discussion

First Paragraph of Discussion could be omitted because most of it has been mentioned in Introduction and is not a Discussion concerning the Results of the study.  In addition, previous studies mentioned in paragraph 2, need to be incorporated late in Discussion with respect to the Results obtained by the authors in their study.

Line 305: Note Kawakami et al., (2011) used primary cortical neuronal cultures, which are not too dissimilar to the cortical neuronal cultures used by the authors in their study.  Consider rewording sentence.

Paragraph 3: Too much rehashing of results and needs to be improved.  Discussion should be discussing implications/significance of results, and how findings relate to previous studies if applicable. Also, if possible, have a separate paragraph Discussing different parameters measured.

Line 314/315: “We have shown that ILG significantly increase lag-period in cell culture with Glu administration (Figure 2).”  I am struggling to see data/graph that shows this???

Paragraph 4: Simplify and improve.

Paragraphs 5/6: Not sure if Discussion on NOX, xanthine oxidase and monoamine oxidase and relevant to this study. ROS generation was not examined in the study.

Line 395: delete “of the cerebral cortex of rats”

Author Response

We are grateful to Reviewer for important comments and valuable advice!

Please note that the line numbers have changed after the corrections.

Overall, written quality of manuscript could be improved with copyediting to improving wording and sentence structure.

The manuscript was edited by Dr. Anna P. Berbenyuk (Sechenov First Moscow State Medical University (Sechenov University), 119146, Moscow, Russia. [email protected]

 Can I suggest changing “Glu” to “glutamate” throughout text in manuscript.

“Glu” have been changed to “glutamate” throughout text, except Figures. 

Discussion also needs to be improved and simplified to better explain the implications of the results obtained in the study.

Thank You for this important note! Discussion was improved according to the reviewer’s comments.

Line 132:  Was ILG present during the 1-hour exposure of neurons to glutamate?  Please provide more details regarding glutamate exposure protocol.

Yes, You are right. ILG was presented during the 1-hour exposure of glutamate. We added “and its exposure was maintained during the rest of the experiment” (line 132) for better understanding. We have also added schemes of experiments with sequence of drug additions and timeline in Figures.

Line 176: please provide details of assay medium. In Culture Oxygen Consumption Rate (OCR) studies were neuronal cultures pre-treated with ILG for 1-hour before glutamate exposure?  

Assay medium composition is mentioned in Materials and Methods (Line 165-166) “assay medium, consisting of 130 mM NaCl, 5 mM KCl, 2 mM CaCl2, 1 mM MgCl2, 20 mM HEPES, 5 mM Glucose, at pH ∼7.4”.

Please better describe sequence of substance additions and time frames for OCR studies.  A diagrammatic representation of timeline procedures/steps for OCR studies could be useful.

We added a scheme of experiments (Figure 3C) showing the time protocol of experiment. Thanks for the valuable advice!

Why was glutamate concentration reduced to 30µM in the OCR studies? 

We have previously done similar experiments with 100 µM and this study was no exception. Thank You for pointing out our mistake.

Line 205: change “0,5” to “0.5”.  Please use decimal point rather than “,” throughout manuscript.

Points have been changed to “,” in value throughout manuscript.

Line 206: The 5 µM ILG concentration also appeared to significantly improve neuronal survival after glutamate treatment.

You are right. We forgot to mention it in text. This information was added to the manuscript.

Figure 1: What do 22%, 20% and 13% above bars for 0.5, 1 and 5 µM ILG concentrations represent???

Percentages of neuroprotective effects were removed from the Figure 1 and mentioned in text.

Line 210:  Suggest rewording “a high activity of dehydrogenases” “.. higher WST-test absorbance levels..”

Phrase “a high activity of dehydrogenases” was reworded to “.. higher WST-test absorbance levels..”. (line 214-215)

Line 210/211: “There were no significant differences between the groups.”  What are the authors referring to here???

This sentence was in the wrong place. Thank you for pointing out this inaccuracy. The sentence has been moved up. 

Line 223/224: “Fluorescence microscopy measurements showed that incubation with ILG (1μM, 15 min) did not change [Ca2+]i in resting neurons.”   Where is data/Figure showing this???

Graph of dynamic curves of [Ca2+]i change has been added to the Figure 2A. The graph shows that [Ca2+]i did not change after the addition of ILG until glutamate was used.

Line 224/225: “The addition of Glu (100 μM, 10 μM Gly, 0 Mg2+, 15 min) induced development of delayed calcium deregulation (DCD) and synchronous changes in [Ca2+]i and ΔΨm.”  Where is data/Figure showing this???

Graphs of dynamic curves of synchronous changes in [Ca2+]i and ΔΨm have been added to the Figure 2B-C.

Lines 226/227: Similar synchronism of [Ca2+]i growth and ΔΨm was repeatedly observed under the action of toxic doses of Glu on cultured neurons [8, 11, 37].” Where is data/Figure showing this???  Why are references included here???

We mentioned references, which described in detail the parameters of this experiment.

Figure 2:  Should graphs A-D also show results for glutamate treatment only????  Or is Control the glutamate treatment (if so then control, no glutamate needs to be shown)

In experiments of measuring of [Ca2+]i and ΔΨm in cortical neurons, cell culture with glutamate exposure and without IGL addition was used as a control. Adding only ILG to the cells did not affect the change in these parameters (not shown).

Figure 2: “ B) Integral fluorescent response 232 (Glu). (C) The slopes (tg α) of linear approximations of the signal curve of the potential-sensitive fluorescent dye Fura-FF during application of EGTA solution. (D) The slopes (tg α) of linear approximations of the signal curve of the potential-sensitive fluorescent dye Rh123 during application of EGTA solution.”   Should this be defined in Materials and Methods rather than in Results section.

We would prefer to leave it in the figure caption for a better understanding of the experiment.

Lines 238-239:  Glutamate response with respect to DCD in Figure 2A needs to be shown.  Also, result with respect to DCD (Fig 2A) looks the same for Control and ILG.

Statistical analysis showed difference in the distribution of data, in the absence of a difference in medians, namely: against the background of the action of ILG, the proportion of cells with a high LAG value increased. We have changed the chart type from "box plot" to "Tukey plot" for better visibility.

Lines/paragraph 251-261; Could all this be simplified.  Aren’t results in Fig 2C summarized in the paragraph above??

Paragraph has been simplified.

Also sentence “However, the rate of mitochondrial potential recovery in neurons with ILG 1μM was significantly lower than in neurons of intact cultures (Figure 2D).” be improved.  Do you mean the time for mitochondrial potential recovery significantly shorter???

The rate of recovery of the mitochondrial potential in neurons with ILG 1 µM was significantly lower, i.e. recovery is slower, it means that recovery of the initial level of mitochondrial potential is longer. We have added a clarification to the text (Line 275-276): «It means cells with ILG 1 μM need more time to reach fully recovery of mitochondrial potential.».

Figure 3B: “As shown in Figure 3B, ILG at low doses (0.1–1 μM) did not influence basal oxygen consumption”. This is with glutamate exposure (30µM) correct?, but at what time point after glutamate exposure or duration of glutamate exposure???? Looking at Fig 3B, it would appear 5 µM causes decrease OCR.

We added new Figure, therefore 3B now is 4B. The text of manuscript was changed according to the Figure 4B,C : “As shown in Figure 4B, ILG at low doses (0.1–1 μM) did not influence basal oxygen consumption. The highest dose tested, 5 μM ILG, caused a decrease in basal respiration (Figure 4B) but a slight increase in overall OCR (Figure 4C). Further analysis of the results clarified that this increase in OCR is associated with an increase in non-mitochondrial respiration (Figure 4C).” (Lines 295-299)

Figure 3D:  What was time frame for measurements?

We added the scheme (Figure 3C) showing the time protocol of experiment.

First Paragraph of Discussion could be omitted because most of it has been mentioned in Introduction and is not a Discussion concerning the Results of the study.  In addition, previous studies mentioned in paragraph 2, need to be incorporated late in Discussion with respect to the Results obtained by the authors in their study.

The discussion has been modified as recommended by the reviewer. Results not listed but discussed.

Line 305: Note Kawakami et al., (2011) used primary cortical neuronal cultures, which are not too dissimilar to the cortical neuronal cultures used by the authors in their study.  Consider rewording sentence.

We have modified this and the following sentences.

Paragraph 3: Too much rehashing of results and needs to be improved.  Discussion should be discussing implications/significance of results, and how findings relate to previous studies if applicable.

The discussion has been modified as recommended by the reviewer. Results not listed but discussed.

Also, if possible, have a separate paragraph Discussing different parameters measured.

We added required paragraph (Line 357-361).

Line 314/315: “We have shown that ILG significantly increase lag-period in cell culture with Glu administration (Figure 2).”  I am struggling to see data/graph that shows this???

Statistical analysis showed differences in the distribution of the data form in the absence of a difference in medians, namely, against the background of the action of ILG, the proportion of cells with a large Lag value increased. We have changed the diagram type for better clarity.

Paragraph 4: Simplify and improve.

The paragraph has been improved.

Paragraphs 5/6: Not sure if Discussion on NOX, xanthine oxidase and monoamine oxidase and relevant to this study. ROS generation was not examined in the study.

We agree with the reviewer and therefore we have reduced this section.

Line 395: delete “of the cerebral cortex of rats”

Phrase “of the cerebral cortex of rats” was deleted.

Round 2

Reviewer 1 Report

Authors have addressed most of the issues satisfactorily.

Reviewer 2 Report

The authors have addressed the concerns raised, and therefore the manuscript is worthy for consideration of acceptance.

There are two Figure 2 legends; please correct, and some additional copyediting may be required.